# Genotype × Environment Interaction and Stability Analysis of Selected Cassava Cultivars in South Africa

**DOI:** 10.3390/plants12132490

**Published:** 2023-06-29

**Authors:** Assefa B. Amelework, Michael W. Bairu, Roelene Marx, Mark Laing, Sonja L. Venter

**Affiliations:** 1Agricultural Research Council, Vegetable, Industrial and Medicinal Plants, Private Bag X293, Pretoria 0001, South Africa; bairum@arc.agric.za (M.W.B.); pienaarr@arc.agric.za (R.M.); sventer@arc.agric.za (S.L.V.); 2African Centre for Crop Improvement, School of Agriculture, Earth and Environmental Sciences, University of KwaZulu-Natal, Private Bag X01, Scottsville, Pietermaritzburg 3209, South Africa; 3School of Agricultural Sciences, Food Security and Safety Niche Area, Faculty of Natural & Agricultural Sciences, North-West University, Private Bag X2046, Mmabatho 2735, South Africa; laing@ukzn.ac.za

**Keywords:** AMMI model, genotype × environment interaction, *Manihot esculenta*, stability, breeding sites

## Abstract

Cassava (*Manihot esculenta* Crantz) is an important root crop worldwide. It is adapted to a wide range of environmental conditions, exhibiting differential genotypic responses to varying environmental conditions. The objectives of this study were: (1) to examine the effect of genotype, environment and genotype × environment interaction (GEI) on fresh root yield (FRY) and dry matter content (DMC); and (2) to identify superior genotypes that exhibit high performance for the traits of interest using the genetic tools of additive main effects and multiplicative interaction (AMMI) and genotype stability index (GSI) analysis. Eleven cassava genotypes were evaluated in a randomized complete block design at six trial sites in South Africa. The combined analysis of variance based on AMMI revealed significant genotype, environment and GEI for the traits. The percentage variation due to GEI was higher than the percentage variation due to genotype for FRY, reflecting differential genotypic responses across the experimental sites. The proportion of variance due to genotype variation was larger for DMC. Genotype stability index (GSI) showed that UKF3 (G6), 98/0002 (G2) and P4/10 (G5) were the highest yielding and most stable genotypes for FRY, and 98/0002 (G1), UKF3 (G6) and UKF9 (G11) were the highest yielding and most stable genotypes for DMC. Cultivars 98/0002 and UKF3 were identified as providing high stability with superior fresh root yield and DMC. These genotypes could be recommended to farmers for food, feed and industrial applications without the need for further breeding. The AMMI-2 model clustered the testing environments into three mega-environments based on the winning genotypes for FRY and DMC. Mabuyeni (KwaZulu-Natal), Shatale (Mpumalanga) and Mandlakazi (Limpopo) would be the best testing sites in future cassava-genotype evaluation and breeding programs. This study provides a baseline for a future study on the GEI of cassava varieties, using a larger set of genotypes, factoring in seasonal variation.

## 1. Introduction

With increasing climate variability, cassava (*Manihot esculenta* Crantz) has proven to be among the most resilient food security crops for more than 800 million people in Sub-Saharan Africa. Cassava produces 40% more starch than rice and 25% more than maize [1]. The potential for cassava to play a key role in food security, climate risk mitigation and import substitution for industrial starch, livestock feed and biofuel feedstock in South Africa’s economy has been reviewed by Amelework et al. [2]. Farmers can grow and harvest cassava on marginal soils with minimal capital input and rainfall of less than 500 mm per annum. In South Africa, 2.5 million households practice small-scale subsistence farming on less than 15% of the available agricultural land. The majority of this land is not suitable for the production of maize or vegetable crops, being in low rainfall areas with poor soils. A low-input and rainfed cassava farming system would benefit rural resource-poor farmers in South Africa [3]. However, it has only been grown as a minor subsistence crop by smallholders in the North East regions bordering Mozambique. The ARC and research partners have been researching agronomically suitable cassava genotypes and appropriate production systems in three provinces in South Africa. As a result, many farmers have expressed an interest in farming cassava.

New genotypes of cassava can be developed in a specific research location through either hybridization or mutation. However, to select the best-performing genotypes, it is necessary to evaluate the advanced breeding lines in a wide range of environments. Breeding lines tested in different environments always exhibit significant variation in terms of phenotypic performance owing to environmental variation and different biotic and abiotic stresses [4]. The analysis of genotypic interactions with the environmental conditions of multiple sites helps to quantify the adaptability and stability of genotypes [5]. The differential response of genotypes to different environmental conditions is termed the genotype by environment interaction (GEI).

The presence of GEI is a challenge for breeders in evaluating lines in multilocational trials. Quantifying and minimizing the GEI remains one of the top priorities of any breeding program. GEI reduces the association between phenotypic and genotypic values, thereby hampering the genetic progress in plant breeding programs [6]. The use of statistical models such as additive main effects and multiplicative interaction (AMMI) and genotype (G) main effect plus genotype × environment (GE) interaction (GGE) models assist breeders in quantifying and understanding the patterns of GEI and in evaluating the performance of genotypes in various environmental conditions. This allows breeders to select stable and adaptable genotypes for a range of environments [7].

In Africa, Latin America and Asia, where cassava is grown as a subsistence and industrial crop, research has been conducted to enhance the genetic profile of cassava, resulting in many genotypes being released for improved yield, dry matter and starch content, and resistance or tolerance to major insect pests and disease. Genetic improvement begins with the collection and evaluation of diverse genetic resources [8]. In the past five years, the Agricultural Research Council of South Africa (ARC) has collected a number of cassava cultivars from national and international research institutes. However, the deployment of these newly introduced genotypes into new production areas requires a basic understanding of their performance in the new environments and to identify the most useful environments for future testing and characterization of cassava germplasm [4]. The aims of this study were to evaluate the yield and dry matter content of selected cassava genotypes across different environments, to study the patterns of GEI, to identify superior genotypes that exhibit high-performing and stable genotypes in support of establishing a cassava starch industry in South Africa, and to identify mega-environments for future germplasm evaluation trials.

## 2. Results

### 2.1. AMMI Analysis of G × E Interaction

The AMMI model analysis of variance (ANOVA) of eleven cassava genotypes measured in four environments showed that the mean square (MS) for genotype, environment and GEI were highly significant (*p* < 0.001) for fresh root yield and dry matter content (Table 1). GEI was further partitioned by principal component analysis, which showed that the first two IPCAs MS (IPCA1 and IPCA2) were significant (*p* < 0.001) for both fresh root yield and dry matter content. It was evident from the AMMI analysis that the GEI sum of squares (SS) accounted for a larger proportion of the treatment SS (45.6%) for FRY, whereas the SS of the genotypes constituted the largest proportion of the treatment SS (51.9%) for DMC. The genotype SS accounted for 18.1% of the treatment SS for FRY, whilst environment and GEI SS accounted for 36.4% and 45.6%, respectively (Table 1). Unlike FRY, environment and GEI effects SS accounted for a smaller proportion of the treatment SS for DMC. The model with the first few IPCAs that captured most of the GEI variation was the best model for extracting and explaining the GEI pattern from the dataset. In this study, the percentage goodness of fit by the first two IPCAs was 74.7% for FRY and 77.4% for DMC, indicating the usefulness of the AMMI model for extracting and understanding the patterns of GEI. The results also showed that the six environments varied both in the main and interaction effects.

### 2.2. The GEI Patterns of Traits and Genotypes Based on GGE Biplot Analysis

To visualize the performance of different genotypes across different environments, biplots were used. The AMMI and GGE biplots are generally used to explain the genotype adaptation or stability across environments. If the PCA score for a genotype or environment is near zero, then there is a small interaction impact; however, if a genotype and environment have the same sign on the PCA axis, there is a positive interaction; alternatively, they have different signs on the PCA axis, there is a negative interaction. Environments with large PCA scores show high interaction between the environments and genotypes and are discriminatory, whereas environments with PCA scores near zero have little interaction with the genotypes and have a low discriminatory value.

### 2.3. Fresh Root Yield

The GEI IPCA1 scores were plotted against the mean performances of the genotypes and environments in the AMMI1 model (Figure 1A). The x-coordinate indicates the main effects (means), and the y-coordinate indicates the effects of the interaction (IPCA1). The genotype and environment overall mean was 70.4 tons ha^−1^. The superior genotypes were G1 > G3 > G6 > G7 > G5 > G11 > G2, which were located on the right two quadrants (top and bottom right) of the biplot. Environments were distributed from low-yielding environments in the left two quadrants (top and bottom left) to the high-yielding environments in the right two quadrants (top and bottom right). E3 > E6 > E2 > E4 were identified as higher-yielding environments, whereas E1 and E3 were relatively low-yielding environments.

Values closer to the origin of the axis (IPCA1) provide a smaller contribution to the interaction than those that are further away. Hence, G9 and G11 were the most unstable genotypes, with mean fresh root yields of 65.1 ton ha^−1^ and 71.4 ton ha^−1^, respectively, which are close to the overall mean (Figure 1A), while G2 and G5 were relatively stable genotypes. On the contrary, the majority of the genotypes revealed intermediate stability and performance. However, among these genotypes, 98/0002 had the highest mean fresh root yield (86.8 ton ha^−1^) combined with stability comparable to the other genotypes. Similarly, some environments, such as E2 and E3, stood out as making little or little contribution to the interaction, E6 made a small contribution, and E1, E5 and E4 made a large contribution to the interaction.

The cumulative contribution from IPC1 and IPC2 included about 93% of the interaction MS (Figure 1C). According to the correlation between IPC1 and IPC2, the genotypes that were positioned near the origin had the least interaction, and the genotypes positioned near the axis had a stability that was more general. G1, G7 and G8 showed little or no interaction with the environments, while G2, G4, G5, G6 and G10 revealed a minimum interplay between genotype and environments, whereas G3, G9 and G11 were the most unstable. G1 had a PCA1 score of approximately zero on the IPCA1 axis, indicating that this genotype was the most stable across environments.

E2 and E3 were the largest contributors to the phenotypic stability of the genotypes, and these environments were among the lowest in mean FRY. However, E1, E5 and E6 contributed the most GEI. Any genotypes positioned closer to a certain environment have specific stability in that environment. Hence, G3 had a specific adaptation to E6, G9 to E5 and G5 and G10 to E4.

The partitioning of GGE showed that IPCA1 and IPCA2 accounted for 39.1% and 30.2% of the GGE sum of squares, respectively, explaining a total of 69.3% variation in FRY (Figure 1D). Based on the predicted means of FRY obtained from the AMMI2 model, three mega-environments were identified (Figure 1D). The first one contains E1 and E4, with G7 and G11 as the winner genotypes. The second mega environment constituted E2, E3 and E6, where G1 and G3 were the best genotypes at these sites. The last mega-environment was formed by one environment, E5, where genotype G9 was the winner (Figure 1D).

### 2.4. Dry Matter Content

The scatterplot of mean DMC vs. IPCA1 (Figure 2A) illustrates that G11 (49.9%) and G4 (49.8%) had the highest DMC, while G8 (27.5%) had the lowest. The vertical line that divides the horizontal axis into two parts is the mean DMC (43.3%). G11 > G4 > G6 > G1 > G5 > G2 > G9 had a higher DMC than the mean DMC, while G3, G7 and G8 had lower DMC values. In terms of IPCA1, G10, G2 and G9 had a maximum GEI and were the most unstable genotypes. The highest DMC was recorded at E4, followed by E5 and E6. However, in the rest of the locations, the genotypes performed below the mean DMC. E4 and E2 made large contributions to the GEI, while E1 made a smaller contribution (Figure 2A).

The cumulative percentage of the GEI that was captured by IPCA1 and IPCA2 was 77.4% (Figure 2C). According to the association between IPCA1 and IPCA2, G1, G3, G5 and G6 were the most stable genotypes with the least interaction, while G9, G2, G7, G4, G8 and G11 were the most unstable genotypes. The locations were ranked as E4 > E5 > E6 in terms of DMC. Furthermore, there was a positive interaction between E4 and G2, E2 and G4, E1 and G9.

The partitioning of GGE showed that IPCA1 and IPCA2 accounted for 68.1% and 16.5% of the GGE sum of squares, respectively, explaining 84.6% of the total variation (Figure 2D). Two mega environments were formed based on winning genotypes. The first mega environment constitutes E1, E2, E3, E5 and E6 with G1, G4, G5, G6 and G11 as winning genotypes, whereas the second environment contains E4 with G2, G9 and G10 as the winning genotypes.

### 2.5. Stability Analysis Using AMMI Model

AMMI stability value (ASV) was proposed by Purchase et al. [9] to quantify and rank genotypes according to their stability. The ranking of genotypes based on ASV for FRY and DMC are presented in Table 2 and Table 3, respectively. The genotypes were ranked based on the ASV score, where low scores represent the most stable genotypes. Based on the ASV, the most stable genotype for fresh root yield were G6, G8, G4 and G2, with the lowest AS scores. With regard to DMC, G1, G5, G6 and G3 had the lowest ASV rank and most stable genotypes, while UKF8, G10, G2 and G4 were the least stable.

Another approach to determine yield stability is the use of the genotype stability index (GSI), calculated by ranking the mean performance of genotypes (RY) across environments. The YSI incorporates both mean yield and stability in a single criterion. The YSI ranked G6, G1 and G5 as the highest-yielding and most stable genotypes for FRY, whereas G9, G11 and G10 were the least stable for FRY. On the other hand, G1, G6 and G11 were ranked the highest and the most stable genotypes for DMC, whereas G10, G8 and G2 were ranked the least stable for DMC.

## 3. Discussion

Significant genotype × environment interactions reduce the progress of genotype selection because large interactions can reduce gains from selection and make the identification of superior genotypes difficult. Quantifying and understanding GEI is important when selecting genotypes adapted to a range of target environments that vary considerably [10]. Evaluation of genotypes across different environments is important to identify stable genotypes and high-yielding genotypes in specific environments and to identify sites that best represent the target environment [11]. In addition, genotypic stability and adaptability should be considered important aspects of yield trials [12]. An ideal genotype should have a superior and stable performance within and across environments. Several statistical approaches to the measurement of the stability of performance have been suggested to examine the stability of individual genotypes across environments [13]. Therefore, it is recommended that yield and stability are evaluated simultaneously in multi-site trials to reduce the effect of genotype by environment interaction and to make selection more precise [14].

Genotype effects were highly significant (*p* < 0.001) for FRY and DMC, indicating the presence of wide genetic variation among the genotypes for the traits. This variation suggests that the studied genotypes constituted diverse germplasm with sufficient genetic variation for breeding purposes, which could be improved by hybridization among the genotypes followed by selection. The significant environmental effect (*p* < 0.001) observed for all the agronomic traits signified the substantial influence of the environment on the expression of the traits. This variation underlines the need to conduct multi-locational trials in order to identify genotypes with broad or specific adaption. The significant variation of the GEI effect (*p* < 0.001) found for the observed agronomic traits indicated that genotype and environment main effects were not sufficient to explain the observed phenotypic variation. This variation was the source of deviation in the performance of the genotypes across the different environments. Many researchers have reported similar findings [15,16,17,18,19,20]. The results confirmed that testing for GEI and assessing the stability of genotypes across environments is essential in breeding programs.

The AMMI analysis of variance revealed that GEI contributed the most variability (45.6%) of the total variation for the parameter FRY. However, the genotypic variance accounted for a large proportion of the observed phenotypic variance (51.9%) for DMC. Environmental variation contributed to 36.4% of the total variability in FRY and 21.3% in DMC. As such, only 18% of the variation was explained by the genetic contribution for the parameter FRY. Olayinka et al. [20] in Nigeria reported that in their study on cassava, more than 88% of the treatment SS was due to environmental variation for FRY. In this study, the variation in GEI constituted the larger proportion of the treatment SS for FRY. This finding agreed with many researchers [16,17,18,19,21,22,23], who reported the significant influence of the environment and GEI on the expression of yield and yield component traits. However, this finding was contrary to Benesi et al. [24], Olayinka et al. [20], Peprah et al. [25] and Tumuhimbise et al. [26], who observed non-significant GEI values for DMC, FRY and starch. The discrepancies could be due to the fact that genotypes might have similar responses across environments or that the testing environments that they used were similar in terms of spatial and temporal environmental conditions. The results of this particular study showed that the selected environments were adequately diverse to discriminate between genotypic performances under the different temporal and spatial environmental conditions used [27].

Stability analysis methods are often used to identify genotypes that have stable performance and respond positively to improvements in environmental conditions [28]. AMMI stability value (ASV) indicates the stability of genotypes in which genotypes with low ASVs are considered to be stable, whilst those with high ASV values are considered to be less stable genotypes [29]. Cultivars UKF3 (G6), UKF5 (G8) and P1/19 (G4) were the most stable for FRY and 98/0002 (G1), P4/10 (G5) and UKF3 (G6) were the most stable for DMC. Genotypic performance *per se* can be misleading due to the sensitivity of genotypes to environmental fluctuations. Similarly, stability alone does not ensure a high yield since a consistently low-yielding genotype can also be stable [30]. In some cases, the most stable genotypes do not have the best yield performances [17]. Hence, for breeding, agronomy and physiological studies, both performance and stability should be considered simultaneously to reduce the effect of GEI. Therefore, a high root yield is considered with stability in the estimation of the genotype stability index (GSI). Genotypes with lower GSI are desirable because they combine a high mean yield performance with stability [26]. Based on the GSI, UKF3 (G6), 98/0002 (G1) and P4/10 (G5) were identified as providing both high yield performance and stability for FRY, and 98/0002 (G1), UKF3 (G6) and UKF9 (G11) were identified as having high DMC and stability values.

The results of AMMI analysis indicated that the first two IPCAs were highly significant (*p* < 0.001) and contributed 74.7% and 77.4% of the total phenotypic variation for FRY and DMC, respectively. Gauch [31] proposed that the most accurate model for AMMI could be predicted using the first two IPCAs; hence the two IPCA scores were then used in the calculation of ASV, as postulated by Purchase et al. [9]. Genotypes with IPCA1 scores adjacent to the zero lines of the biplot indicated that these genotypes are suited to all environments, whereas IPCA1 vectors with the same sign and score but which are situated away from the zero lines of the biplot have genotypes that are adapted to a specific environment [32]. MSAF2 (G3) had IPCA1 scores of close to zero on the IPCA1 axis, which indicated that this genotype was suitable and stable across environments for FRY, while 98/0002 (G1), P4/10 (G5) and UKF4 (G7) were suitable and stable for DMC. The test sites Masibekela and Mabuyeni generated low interaction effects for FRY, while the sites Mandlakazi and Masibekela generated low interaction effects for DMC.

Studies on the suitability of the test sites to reflect the target environments is the next important step after determining GEI to quantify each of the test environments for their discriminative power, representativeness, inter-relatedness and redundancy among the test environments [33]. The concept of a mega-environment was introduced to subdivide a crop production region into several relatively homogeneous mega-environments, to breed and target adapted genotypes for each mega-environment, and to reduce research costs by eliminating redundant trial sites [34]. The first mega-environment recommended by the GGE plot includes Shatale (E4; Mpumalanga) and Nseleni (E1; KZN). Shatale (E4) would be a good environment for future preliminary screening and for breeding activities because it represents an intermediate-performance environment for FRY (71.1 ton ha^−1^) and a high-performance environment for DMC (50.3%). The second mega-environment consisted of Masibekela (E3; Mpumalanga), Mutale (E6; Limpopo) and Mabuyeni (E2; KZN). Mabuyeni (E2) would be a good testing location because it provides a high-performance environment for FRY (78.3 ton-ha^−1^) and an intermediate-performance environment for DMC (40.1%). The high FRY and DMC observed in Mutale could be attributed to the crop being harvested more than 2 months later than at the other sites. The third mega-environment only contained Mandlakazi (E5) (Limpopo). The three mega-environments represented the three provinces (Mpumalanga, KZN and Limpopo) in South Africa that are suitable for cassava cultivation.

The specific adaptability of a genotype to a particular environment could be assessed by analyzing the position of the genotypes with reference to the environmental vectors in the AMMI2 biplot. In addition, a “which-won-where” biplot was constructed for each trait to explore the possible existence of mega-environments within the studied environments and to identify winning genotypes in each mega-environment using GGE analysis. Scavo et al. [35], studying potato genotypes, and Khan [36], studying Bambara groundnut genotypes, used a similar approach. The results confirm the presence of distinct interactions between genotypes and environments for FRY and DMC. In the biplot, UKF9 (G11) and UKF4 (G7) were relatively far from the origin of the axes and close to Shatale (E4), indicating that they are well adapted for Shatale for FRY. On the other hand, 98/0002(G1), MSAF2 (G3) and UKF3 (G6) were found to be the best in the second mega-environment, Mutale (E6), with above-average values, while UKF7 (G9) was found to be well adapted to Mandlakazi. For DMC, only two mega-environments were identified, in which the first environment comprised all the environments except Mandlakazi with 98/0002 (G1), P1/19 (G4), P4/10 (G4), UKF3 (G6) and UKF9 (G11) as winning genotypes, and the second mega-environment consisted of only one location, Mandlakazi (E4), with 98/0505 (G2), UKF8 (G10) and UKF7 (G9) as the winning genotypes.

## 4. Materials and Methods

### 4.1. Trial Site Description

This research was conducted in three provinces that represent the tropical and subtropical agroecological zones in South Africa: KwaZulu-Natal (KZN), Mpumalanga and Limpopo. The data presented in this study were collected from 6 environments, namely Nseleni, Mabuyeni, Masibekela, Shatale, Mutale and Mandlakazi, during the 2019–2020 cropping season. Cassava yield and yield component traits were evaluated approximately 12 months after planting (MAP) at all locations except in Mutale (14 months). Detailed information on the location of each trial site and their GPS coordinates are presented in Table 4.

### 4.2. Planting Material and Experimental Design

The study evaluated 11 cassava genotypes acquired from the International Institute of Tropical Agriculture (IITA), the University of KwaZulu-Natal (UKZN) and the Agricultural Research Council (ARC; Table 5). Planting materials were multiplied using an in vitro tissue culture system, and the plantlets were acclimatized in a greenhouse before planting. The trial at each location was laid out in a randomized complete block design with 3 replications. Each genotype was planted in a plot size of 5 × 5 m comprising 5 rows of 5 plants with an inter- and intra-row spacing of 1 x 1 m, respectively. Plants were grown from disease-free, in vitro tissue cultured plantlets. The genotypes were grown under rainfed conditions. Neither fertilizers nor pesticides were applied. Standard agronomic practices were followed as recommended for cassava [37].

### 4.3. Data Collected and Preparation of Samples

Yield data was collected from five randomly selected plants per plot. The mean yield of the five plants was converted into ton ha^−1^ using a plant density of 10,000 plants per hectare.

Dry matter content (*DMC*) was measured using 5 randomly selected storage roots. The roots were thoroughly cleaned with water and dried with a paper towel before being diced into 1 cm thick discs at 25%, 50% and 75% of the length from the base of the roots. The freshly-cut tuber discs were further sliced into smaller-sized cubes to facilitate oven drying. Five 100 g chopped cubes were taken from each sample and were oven-dried at 105 °C for 24 h. The dried cubes were weighed to obtain the dry matter content.

*DMC* was measured using the following equation:$$Dry\;matter\;content\;\left(DMC\right)=\left(\frac{DW}{FW}\right)\times 100$$

### 4.4. Data Analysis

All the data generated were analyzed using GenStat statistical software version 19.1 [38]. The quality of the data was inspected for data logging errors, and outliers and extreme values were removed from the analysis. Data obtained from each location were analyzed separately by running a single location analysis of variance, and thereafter data from all four environments were pooled for analysis of variance (ANOVA) to perform the combined analysis of 11 cassava genotypes across the four environments to test the presence of significant genotype, environment and genotype × environment variation.

The pooled ANOVA was highly significant (*p* < 0.001) for genotype, environment and GEI components for FRY and *DMC*, justifying the use of Additive Main effect and Multiplicative Interaction (AMMI) and GGE biplot analyses to identify the stable genotypes. The AMMI model was used to determine the main effects and genotype × environment interactions. The AMMI model fits the additive effects for the genotypes and environments and the multiplicative term for interactions [39]. The AMMI model was as follows:$$Y_{ij}=\mu +\alpha_{i}+\beta_{j}+\sum_{k=1}^{n}\lambda_{k}\gamma_{ik}\delta_{jk}+\varepsilon_{ij}$$
where *Y_ij_* = the yield of *i*th genotype in the *j*th environment over all replications, µ is the grand mean, 𝛼I is the *i*th genotype mean deviation, *β_j_* = the *j*th environment mean deviation, *λ_k_* is the singular value for IPC axis *k*, *γ_ik_* is the *i*th genotype eigenvector value for IPC axis *k*, *δ_jk_* is the *j*th environment eigenvector value for IPC axis *k* and *ε_ij_* is the error term.

Biplots were generated by plotting the first principal component axis (IPCA1) scores of the genotypes and the environments against their respective IPCA2 scores, resulting from the singular value decomposition of the environment or standardized G × E data [40]. The genotype main effect and genotype by environment interaction effect (GGE) biplot were generated based on 2 concepts, the biplot concept [41] and the GGE concept [42]. A GGE biplot analysis was applied for visual examination of the GEI pattern in the data set.

An AMMI stability value (*ASV*) was calculated for each genotype according to the relative contributions of the principal component axis scores (*IPCA1* and *IPCA2*) to the interaction sum of squares. The AMMI stability value (*ASV*), as described by Purchase et al. [9], was calculated as follows:$$ASV=\sqrt{{\left[\frac{IPCA1_{SS}}{IPCA2_{SS}}\;\left(IPCA_{score}\right)\right]}^{2}\;{\left(IPCA2_{score}\right)}^{2}}$$

The genotype stability index (YSI) was also calculated using the sum of the ranking based on yield and ranking based on the AMMI stability value.
GSI=RASV+RY
where *RASV* = the rank of the genotypes based on the AMMI stability value; *RY* = the rank of the genotypes based on yield across environments. YSI incorporates both mean yield and stability in a single criterion. Low values of both parameters show desirable genotypes with high mean yields and stability.

## 5. Conclusions

The aim of this study was to evaluate and assess the adaptability and stability of selected cassava genotypes based on their mean performance under a wide range of environments in order to select superior and stable genotypes. Selection of genotypes for stability is needed in most dryland environments, where the environment is variable and unpredictable. The stability and adaptability analysis, using AMMI biplots, ASV and GSI statistics, identified the cassava genotypes UKF3 (G6), 98/0002 (G1) and P4/10 (G5) as being the most stable genotypes with the highest root yields, while 98/0002(G1), UKF3 (G6) and UKF9 (G11) were found to be the most stable genotypes with high DMC. Cultivars 98/0002 and UKF3 were identified as combining high stability with superior FRY and DMC. These genotypes could be recommended to farmers for food, feed and industrial application. In addition, genotypes identified as high performing and stable for both traits could be utilized as parental genotypes in future breeding programs. Similarly, the test sites of Mandlakazi (Limpopo), Mabuyeni (KZN) and Shatale (Mpumalanga) were identified as suitable and representative environments for all genotype evaluations and breeding for FRY and DMC. This study will serve as a baseline for further studies on GEI effects with a larger set of cassava genotypes and will factor in seasonal effects.

## Figures and Tables

**Figure 1 plants-12-02490-f001:**
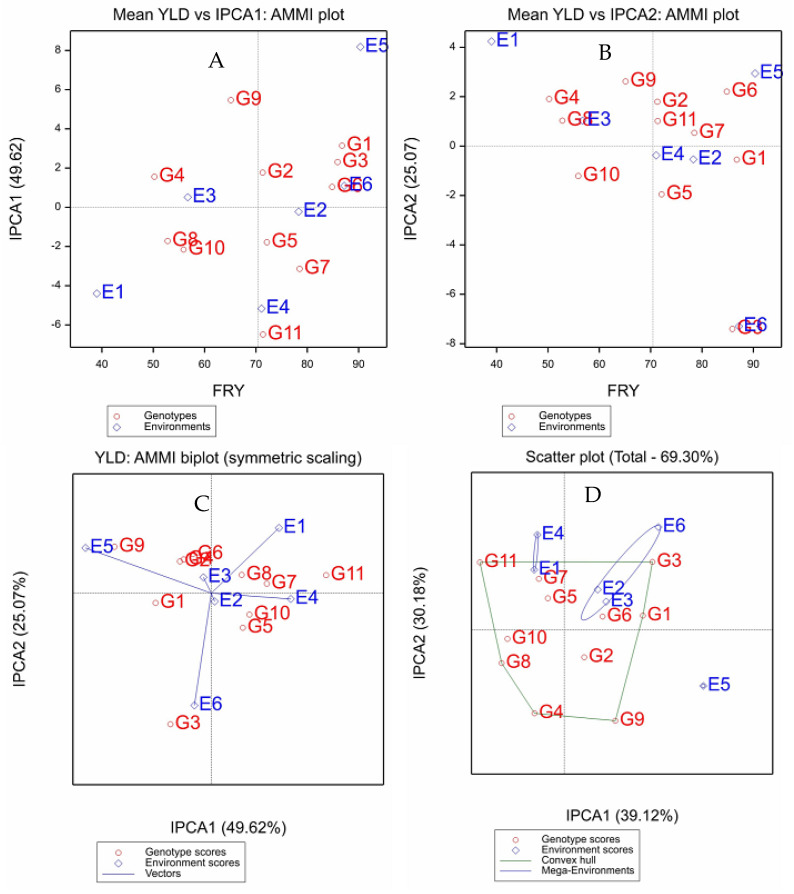
Biplot analysis for 11 cassava genotypes evaluated in six trial sites for fresh root yield (FRY). (**A**) = AMMI1 biplot showing the IPCA1 vs. main effect (means); (**B**) = AMMI1 biplot showing the IPCA2 vs. main effect; (**C**) = AMMI2 biplot showing the first two principal axes of interaction (IPCA2 vs. IPCA1) and (**D**) = GGE plot defining mega-environments using different winning genotypes tested. E1–E6 = the six testing locations, and G1–G11 = the eleven cassava genotypes used for this study.

**Figure 2 plants-12-02490-f002:**
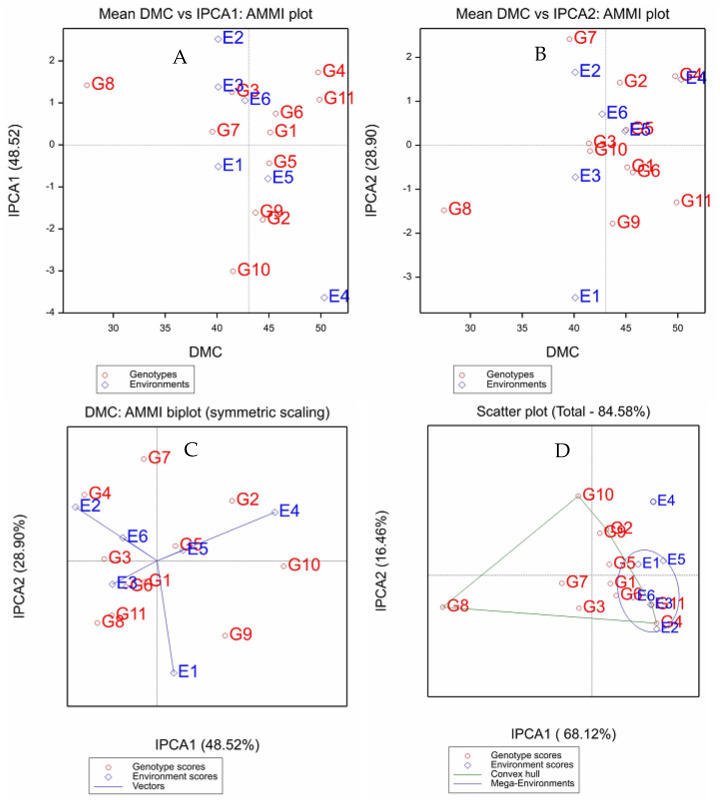
Biplot analysis for 11 cassava genotypes evaluated in six trial sites for dry matter content (DMC). (**A**) = AMMI1 biplot showing the IPCA1 vs. main effect (means); (**B**) = AMMI1 biplot showing the IPCA2 vs. main effect; (**C**) = AMMI2 biplot showing the first two principal axes of interaction (IPCA2 vs. IPCA1); and (**D**) = GGE plot defining mega-environments using different winning genotypes tested. E1–E6 = the six testing locations, and G1–G11 = the eleven cassava genotypes used for this study.

**Table 1 plants-12-02490-t001:** Additive main effects and multiplicative interaction (AMMI) analysis of variance for fresh root yield (t.ha^−1^) and dry matter content (%) of 11 cassava genotypes measured on six trial sites.

Source	DF	Fresh Root Yield			Dry Matter Content		
		SS	MS	% SS Explained	SS	MS	% SS Explained
Treatments	65	173,736	2673		12,784	197	
Genotypes	10	31,367	3137 ***	18.05	6640	664 ***	51.94
Environments	5	63,219	12,644 ***	36.39	2726	545 ***	21.32
Interactions (GEI)	50	79,150	1583 ***	45.56	3418	68 ***	26.74
IPCA 1	14	39,273	2805 ***	49.62	1658	118 ***	48.51
IPCA 2	12	19,846	1654 ***	25.07	988	82 ***	28.91
Residuals	24	20,031	835		772	32	
Error	120	16,722	139		939	7.8	

DF = degree of freedom; IPCA1 = the first interaction principal component; IPCA2 = the second interaction principal component; *** significant at *p* < 0.001.

**Table 2 plants-12-02490-t002:** Ranking of 11 cassava genotypes based on fresh root yield, Additive main effects and multiplicative interaction (AMMI) stability value (ASV) and genotype stability index (GSI) based on six trial sites.

Genotype	Mean	RY	IPCA1	IPCA2	ASV	RASV	GSI	RGSI
G1	86.77	1	3.14	−0.55	6.24	8	9	2
G2	71.31	7	1.77	1.79	3.94	4	11	4
G3	85.91	2	2.31	−7.40	8.70	9	11	6
G4	50.22	11	1.56	1.91	3.63	3	14	8
G5	72.13	5	−1.78	−1.95	4.03	5	10	3
G6	84.84	3	1.04	2.21	3.02	1	4	1
G7	78.51	4	−3.14	0.54	6.23	7	11	5
G8	52.8	10	−1.72	1.03	3.56	2	12	7
G9	65.11	8	5.47	2.62	11.13	10	18	11
G10	55.91	9	−2.15	−1.21	4.43	6	15	9
G11	71.37	6	−6.49	1.02	12.88	11	17	10

RY = mean yield; IPCA1 = First interaction principal component; IPCA2 = Second interaction principal component; ASV = AMMI stability value; RASV = Rank of AMMI stability value; YSI = Genotype stability index; RYSI = Rank of genotype stability index.

**Table 3 plants-12-02490-t003:** Ranking of 11 cassava genotypes based on dry matter content (DMC), additive main effects and multiplicative interaction (AMMI) stability value (ASV) and genotype stability index (GSI) based on their performance at six trial sites.

Genotype	Mean	RY	IPCA1	IPCA2	ASV	RASV	GSI	RGSI
G1	45.13	4	0.30	−0.50	0.71	1	5	1
G2	44.41	6	−1.78	1.43	3.31	10	16	9
G3	41.44	9	1.26	0.04	2.11	4	13	6
G4	49.75	2	1.73	1.57	3.30	9	11	5
G5	45.03	5	−0.44	0.35	0.81	2	7	4
G6	45.66	3	0.74	−0.62	1.39	3	6	2
G7	39.56	10	0.32	2.41	2.47	6	16	8
G8	27.46	11	1.42	−1.48	2.80	7	18	10
G9	43.71	7	−1.62	−1.78	3.25	8	15	7
G10	41.54	8	−3.01	−0.13	5.05	11	19	11
G11	49.88	1	1.08	−1.30	2.23	5	6	3

RY = mean dry matter content; IPCA1 = First interaction principal component; IPCA2 = Second interaction principal component; ASV = AMMI stability value; RASV = Rank of AMMI stability value; YSI = Genotype stability index; RGSI = Rank of genotype stability index.

**Table 4 plants-12-02490-t004:** Details and GPS coordinates of the six trial sites.

Code	Location	District	Province	Soil Type	GPS Coordinate
E1	Nseleni	Empangeni	KZN	Sandy	−28.634120, 31.912331
E2	Mabuyeni	King Cetshwayo	KZN	Silt	−28.853811, 31.961901
E3	Masibekela	Ehlanzeni	Mpumalanga	Sandy loam	−25.870814, 31.825738
E4	Shatale	Ehlanzeni	Mpumalanga	Sandy loam	−24.747785, 31.035320
E5	Mandlakazi	Mopani	Limpopo	Sandy loam	−23.801784, 30.377987
E6	Mutale	Vhembe	Limpopo	Clay	−22.721418, 30.572238

KZN: KwaZulu-Natal.

**Table 5 plants-12-02490-t005:** Descriptions of cassava genotypes tested at six different trial sites.

Code		Type	Source	Trait
G1	98/0002	Released cultivar	IITA	CMD resistance
G2	98/0505	Released cultivar	IITA	CMD resistance
G3	MSFA2	Landrace	ARC	High FRY/Low *DMC*
G4	P1/19	Breeding line	IITA	High *DMC*
G5	P4/10	Breeding line	IITA	High *DMC*
G6	UKF3	Breeding line	Kenya	High SC
G7	UKF4	Breeding line	Kenya	High SC
G8	UKF5	Breeding line	Kenya	High SC
G9	UKF7	Breeding line	Kenya	High SC
G10	UKF8	Breeding line	Kenya	High SC
G11	UKF9	Breeding line	Kenya	High SC

IITA = The International Institute of Tropical Agriculture; UKZN= University of KwaZulu-Natal; ARC= Agricultural Research Council; CMD = cassava mosaic disease; DMC = dry matter content; FRY = fresh root yield; SC = starch content. The UKF materials were originally from Kenya and were used for Ph.D. studies at the University of Kwazulu-Natal. UKZN is the custodian of the materials.

## Data Availability

Not applicable.

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
