# Peer review of "Genotype × Environment Interaction and Stability Analysis of Selected Cassava Cultivars in South Africa"

_plants, 2023, doi:10.3390/plants12132490_

Round 1
Reviewer 1 Report (Previous Reviewer 2)
I just sugest some more corections.
L17. Explain FRY
L17-18. No need to repeat FRY and DMC
L20. Replace environments with testing sites
L. 22 indicating the influence of GEI on the expression of the traits. Replace with . indicating the differential cultivar response across experimental sites.
L 23. The proportion due to the genotype variation was larger for DRC. According to…. replace with. Yield stability index (YSI) shown that, P4/10,……were the highest yielding and the most stable genotypes for FRY……………
L25. and UKF4 were identified ………….and UKF4 was identified……..
L26. Cultivars P4/10 was identified, replace with, Cultivar P4/10 was identified………
Extensive editing of English language required
Author Response
Thank you for the corrections
L17. Explain FRY
- Corrected
L17-18. No need to repeat FRY and DMC
- Corrected
L20. Replace environments with testing sites
- Corrected
- 22 indicating the influence of GEI on the expression of the traits. Replace with . indicating the differential cultivar response across experimental sites.
- Corrected
L 23. The proportion due to the genotype variation was larger for DRC. According to…. replace with. Yield stability index (YSI) shown that, P4/10,……were the highest yielding and the most stable genotypes for FRY……………
- Corrected
L25. and UKF4 were identified ………….and UKF4 was identified……..
- Corrected
L26. Cultivars P4/10 was identified, replace with, Cultivar P4/10 was identified………
- Corrected
Reviewer 2 Report (Previous Reviewer 3)
With regret, I observed that the authors did not accept most of my suggestions. Please see below:
1) line 37: the introduction is always separated from abstract concerning the writing of binomial names. Thereofre, please add here the full binomial name
2) the authors still continue in not follow the journal's guidelines. According to journal requirement, paragraphs and sub-paragraphs must be numbered
3) line 352 and 355: were collected and were excluded
4) lines 354-359: this explanation is not required for the article. Readers want to know what the authors did and how they performed their experiment in order to repeat it. Reporting why they did not consider other locations is not useful and it not justifies that only four locations are not strong enough for GEI analysis
5) Figure 3: I still repeat that it is not useful and perhaps it is low readable. A table with the coordinates and specific weather conditions is much more better. Furthermore, the lack of any information about soil properties is an important deficiency that make not very effective the comparisons between locations
6) Table 4: please spell-out all acronyms in the Table captions
7) paragraph "data analysis" is written with different font and size of text
8) lines 410-412: it is not clear. Please repgrase it. GGE is another type of GEI analysis, different from AMMI. In this study, the authors showed only one type of GGE biplot, i.e. the polygon view. From materials and methos, this is not clear
9) Table 1: the caption of Tables must be provided before the table according to journal requirements
10) Table 1: I still not understand what the "treatment" is. This study did not consider any treatment, but only the genotype, locations and genotype x location effect. In all such kinds of studies the tables are reported in this way. Therefore, please delete the "treatment" row since it creates only confusion
11) Please add in M&Ms the sofwtare used for statistical analysis
12) There are many ways to improve the resolution of Figures, such as programs and online converters
13) In addition to the previous comments, the most important deficiency still remains the weakness of the dataset. I understand the response provided by authors. However, in my opinion only 4 locations and just one growing season (therefore 4 environments) is not enough for a GEI study.
Moderate editing of English language
Author Response
1). line 37: the introduction is always separated from abstract concerning the writing of binomial names. Thereofre, please add here the full binomial name
- The scientific name for cassava is added in the introduction section
1). the authors still continue in not follow the journal's guidelines. According to journal requirement, paragraphs and sub-paragraphs must be numbered
- Apologies for the repeated mistake. The paragraph formatting is corrected according to the journal requirement
3). line 352 and 355: were collected and were excluded
- This comment is not clear.
4). lines 354-359: this explanation is not required for the article. Readers want to know what the authors did and how they performed their experiment in order to repeat it. Reporting why they did not consider other locations is not useful and it not justifies that only four locations are not strong enough for GEI analysis.
- The two location data was included, and data from six locations was reanalyzed and all the figures and tables and the text has been amended accordingly
5). Figure 3: I still repeat that it is not useful and perhaps it is low readable. A table with the coordinates and specific weather conditions is much more better. Furthermore, the lack of any information about soil properties is an important deficiency that make not very effective the comparisons between locations
- This is a participatory evaluation done on-farmers field and we have not done any soil analysis to include in the article. With regard to the question related to the specific weather conditions, all the locations are hot and dry environments but since the experiment was done on-farm, and there was no weather station close by, we do not have specific weather data.
- Figure 3 is replace with Table 4
6). Table 4: please spell-out all acronyms in the Table captions
- Corrected as suggested
7). paragraph "data analysis" is written with different font and size of text
- Corrected
8). lines 410-412: it is not clear. Please repgrase it. GGE is another type of GEI analysis, different from AMMI. In this study, the authors showed only one type of GGE biplot, i.e. the polygon view. From materials and methos, this is not clear
- The difference between GEI and GGE is that in the GGE, the environment components (E) of the GEI is omitted.
9) Table 1: the caption of Tables must be provided before the table according to journal requirements
- Corrected
10). Table 1: I still not understand what the "treatment" is. This study did not consider any treatment, but only the genotype, locations and genotype x location effect. In all such kinds of studies the tables are reported in this way. Therefore, please delete the "treatment" row since it creates only confusion
- It is the AMMI model term that considers genotype and environments as treatments. You can see in all the AMMI model analysis reports.
11). Please add in M&Ms the sofwtare used for statistical analysis
- This information is already included in the MM section on line 397
12). There are many ways to improve the resolution of Figures, such as programs and online converters
- The figure quality was improved. Thank so much you for the lesson.
13) In addition to the previous comments, the most important deficiency still remains the weakness of the dataset. I understand the response provided by authors. However, in my opinion only 4 locations and just one growing season (therefore 4 environments) is not enough for a GEI study.
- The comments taken and two additional environment data was added.
Round 2
Reviewer 2 Report (Previous Reviewer 3)
The authors now addressed my major concerns and consequently I do not have furhter comments. I hope my suggestions improved the overall quality of the manuscript
A language minor revision is required.
Author Response
Thank you.
This manuscript is a resubmission of an earlier submission. The following is a list of the peer review reports and author responses from that submission.
Round 1
Reviewer 1 Report
The paper "Genotype x environment interaction and stability analysis for fresh and dry root yield in selected cassava cultivars in South Africa" corresponds in 1-year evaluation of 11 Cassava cultivars in four locations of South Africa. First of all, with 1-year data is not possible to study GEI since year effect is usually very strong and particularly in a non-irrigated low input crop as the one you tested. Secondly is not possible to investigate the existence of mega-environments with one year data.
The AMMI model was used to study GEI, however it is needed mean caparisons for the parameters. The AMMI just ranks genotypes and we did not know what was the yield difference between genotypes.
Introduction was poor as well results and discussion. There were mistakes in the numbering of Figures. Where are Tables 4,5 and 6?
Generally, this paper need much more work to become published. I suggest conducting trials one more year If your intention is really to study GEI on this crop.
Needs improvment
Author Response
The paper "Genotype x environment interaction and stability analysis for fresh and dry root yield in selected cassava cultivars in South Africa" corresponds in 1-year evaluation of 11 Cassava cultivars in four locations of South Africa. First of all, with 1-year data is not possible to study GEI since year effect is usually very strong and particularly in a non-irrigated low input crop as the one you tested. Secondly is not possible to investigate the existence of mega-environments with one year data.
This project was conducted in 16 on-farm sites, which represent the tropical and sub-tropical agroecological zone that is suitable for cassava cultivation. However, six farmers were eliminated from the project for not following the recommended agronomic practice. Due to planting material shortage, different sets of cultivars were planted at different sites. Only on the six sites that all the 11 cultivars were planted. Out of the six sites, the data collected from two sites at Limpopo and KualZulu-Natal were excluded from the analysis because the trial at Limpopo was planted three months before the rest of the environments and the trial at KualZulu-Natal was miss managed (high weed infestation and the soil was not suitable for cassava). As a result, the FRY and DRC at the two sites were extremely high and low, respectively. Cassava is a biannual crop that can grow from 8 to 24 months. In our case, the data reported in this particular article was collected 12 months after planting, which tells us that the plants had experienced both wet and dry seasons. In the case of cassava, there are many papers G x E published from one-year data for fresh and dry weight, starch content, cassava mosaic disease (CMD), cassava green mite and etc.
The AMMI model was used to study GEI, however it is needed mean caparisons for the parameters. The AMMI just ranks genotypes and we did not know what was the yield difference between genotypes.
Table 2 and 3 show the mean and their ranking for the traits of interest
Introduction was poor as well results and discussion. There were mistakes in the numbering of Figures. Where are Tables 4,5 and 6?
A paragraph has been included on the introduction (44-61). The numbering of figure has been corrected and Table 2 and 3 has been included.
Generally, this paper need much more work to become published. I suggest conducting trials one more year If your intention is really to study GEI on this crop.
The project is completed but we recommended to study the patterns of GEI with a large set of germplasm and include the year (seasonal) variation to see if it has a significant effect.
Reviewer 2 Report
1. Only two traits were apparently considered for this study (FRY and DMC) as described in the Abstract and M&M section. However, the first paragraph of Result section (Table 2) talks about several trait TSC, AMY and AMYP which are not shown in Table 2. Please present these results in Table 2 or remove them from the text.
2. Unlike Table 2, Table 3 includes results from many more traits which are not described in M&M section. The focus of this study seems to be on FRY and DMC, so it is confusing to see inconsistency between M&M section and Results section. Please check and make changes.
3. Table 1 title say that 11 cultivars were tested at 6 locations, but the text at different places suggest that the data was collected from 4 locations.
4. Table 2: The SS values were apparently used to calculate the relative contribution of difference sources of variation to the total variance. For example, the statement “The genotype SS accounted for 23.8% of the treatment SS for FRY, whilst environment and GEI SS accounted for 22.7% and 53.5%, respectively”. I can’t check how these values were derived from the data presented in Table 2? Authors would need to present the data used for these calculations.
5. Nonetheless, in my view the approach (i.e., using SS values from the AVOVA) used in this study to check the relative contribution of different factors is not correct for this purpose. The variance components for each factor (e.g., G, E, GEI) are first to be estimated from the ANOVA by equating the MS to the expected mean squares (EMS). Then, you can calculate the percentage of variance attributed to each factor in the model.
6. I suggest authors to reanalyse the data using a combination of AMMI and REML approach for the mixed model analysis in order to calculate the variance of different effects in the model. Here is a reference for details of this approach: Adjebeng-Danquah J, Manu-Aduening J, Gracen VE, Asante IK, Offei SK. AMMI stability analysis and estimation of genetic parameters for growth and yield components in cassava in the forest and guinea savannah ecologies of Ghana. International Journal of Agronomy. 2017 Jan 1;2017.
Author Response
- Only two traits were apparently considered for this study (FRY and DMC) as described in the Abstract and M&M section. However, the first paragraph of Result section (Table 2) talks about several trait TSC, AMY and AMYP which are not shown in Table 2. Please present these results in Table 2 or remove them from the text.
Table 1 has been modified and the other traits were deleted
- Unlike Table 2, Table 3 includes results from many more traits which are not described in M&M section. The focus of this study seems to be on FRY and DMC, so it is confusing to see inconsistency between M&M section and Results section. Please check and make changes.
The confusion has been addressed and the focus of this article is FRY and DMC
- Table 1 title say that 11 cultivars were tested at 6 locations, but the text at different places suggest that the data was collected from 4 locations.
Explanation has been added on the M&M (lines 347-356)
- Table 2: The SS values were apparently used to calculate the relative contribution of difference sources of variation to the total variance. For example, the statement “The genotype SS accounted for 23.8% of the treatment SS for FRY, whilst environment and GEI SS accounted for 22.7% and 53.5%, respectively”. I can’t check how these values were derived from the data presented in Table 2? Authors would need to present the data used for these calculations.
Table 1 and Table 2 have been merged to avoid confusion. The percentage contribution of genotypes to the total phenotypic variation was calculated as follows.
% SS G = (SS for G/ SS for treatment)*100 = (20455/86031)*100 = 23.8%
- Nonetheless, in my view the approach (i.e., using SS values from the AVOVA) used in this study to check the relative contribution of different factors is not correct for this purpose. The variance components for each factor (e.g., G, E, GEI) are first to be estimated from the ANOVA by equating the MS to the expected mean squares (EMS). Then, you can calculate the percentage of variance attributed to each factor in the model.
The objective of the study is to analyze the performance and adaptability of the exotic cassava cultivars across different environments in South Africa and to select superior and stable genotypes for each environment or across environments. A combined analysis of variance was done and all the variance components were found to be significant. This model does not adequately describe the patterns of GEI, hence we employed AMMI and GGE models to further elucidate the patterns of GEI.
- I suggest authors to reanalyse the data using a combination of AMMI and REML approach for the mixed model analysis in order to calculate the variance of different effects in the model. Here is a reference for details of this approach: Adjebeng-Danquah J, Manu-Aduening J, Gracen VE, Asante IK, Offei SK. AMMI stability analysis and estimation of genetic parameters for growth and yield components in cassava in the forest and guinea savannah ecologies of Ghana. International Journal of Agronomy. 2017 Jan 1;2017.
In the above-mentioned article, the author calculated the variance components (genotypic, phenotypic, and GxE) to calculate the broad sense heritability, which they used to calculate the genetic advance. However, they used the AMMI model and genotype stability index (GSI) to select superior and stable genotypes the way we did in the current article.
Reviewer 3 Report
The manuscript plants-2367701 is an interesting paper about the GEI in cassava. Please find below some comments:
1) line 48: please spell-out the acronyms on their first appearance;
2) line 52: please add the cassava binomial name in full, since it is its first appearance in the text. I suggest also introducing better the crop of this paper with further information
3) Please add the numbers in paragraphs and sub-paragraph titles
4) lines 320-326: I suggest deleting Figure 1 and adding a Table with the geographical coordinates and detailed soil characteristics (% of textural classes, organic matter content, amount of N, P and K, etc.) and weather conditions (monthly rainfall, minimum and maximum air temperatures) for each location. This information is of key importance in such kinds of studies
5) Table 1: please delete the column "Remark"
6) The numbering of Tables and Figures is not consistent since Material and Methods is the last paragraph according to the journal's guidelines. Please check it.
7) Table 2: Which are the treatments under study? None information is provided in materials and methods neither in the aims of the introduction. Please specify it and add this information within the text
8) Only four locations and just one year is quite low for a GEI analysis. Data would be more powerful if repeated for another growing season
9) Please improve the methodology about the yield related traits, too low information is provided.
10) Figure 2: please replace "Main effect" with the corresponding specific trait. In addition, pelase add letters (B) and (D)
11) Figure 3: the same as Figure 2. Please delete the space symbol
12) the resolution and quality of Figures is too low. Please chnage with at least a 300 dpi resolution
13) References are quite outdated. The scientific literature shows more recent papers. I suggest https://doi.org/10.3390/agronomy13010101 and https://doi.org/10.1016/j.scienta.2022.111750
Minor editing of English language required
Author Response
1) line 48: please spell-out the acronyms on their first appearance;
The acronym “GGE” has been written in full
2) line 52: please add the cassava binomial name in full, since it is its first appearance in the text. I suggest also introducing better the crop of this paper with further information
The cassava binomial name was mentioned in the abstract and no need to repeat in the introduction.
3) Please add the numbers in paragraphs and sub-paragraph titles
The paragraph formatting was done according to the journal requirement
4) lines 320-326: I suggest deleting Figure 1 and adding a Table with the geographical coordinates and detailed soil characteristics (% of textural classes, organic matter content, amount of N, P and K, etc.) and weather conditions (monthly rainfall, minimum and maximum air temperatures) for each location. This information is of key importance in such kinds of studies
This research was conducted under on-farm conditions and the soil analysis was not done for each environment. However, the authors agreed that it would have been ideal if the results were supported with soil and weather information to further elucidate the cause of the significant variation in genotypic performance and environment variation.
5) Table 1: please delete the column "Remark"
The entire Table is modified
6) The numbering of Tables and Figures is not consistent since Material and Methods is the last paragraph according to the journal's guidelines. Please check it.
The table numbering has been corrected
7) Table 2: Which are the treatments under study? None information is provided in materials and methods neither in the aims of the introduction. Please specify it and add this information within the text
The treatment in our case was mentioned in the current study 11 genotypes were evaluated in four environments. The AMMI model considers these variations as treatment and decomposed it as genotype, environment, and G x E interaction.
8) Only four locations and just one year is quite low for a GEI analysis. Data would be more powerful if repeated for another growing season
This project was conducted in 16 on-farm sites, which represent the tropical and sub-tropical agroecological zone that is suitable for cassava cultivation. However, six farmers were eliminated from the project for not following the recommended agronomic practice. Due to planting material shortage, different sets of cultivars were planted at different sites. Only on the six sites that all the 11 cultivars were planted. Out of the six sites, the data collected from two sites at Limpopo and KualZulu-Natal were excluded from the analysis because the trial at Limpopo was planted three months before the rest of the environments and the trial at KualZulu-Natal was miss managed (high weed infestation and the soil was not suitable for cassava). As a result, the FRY and DRC at the two sites were extremely high and low, respectively. Cassava is a biannual crop that can grow from 8 to 24 months. In our case, the data reported in this particular article was collected 12 months after planting, which tells us that the plants had experienced both wet and dry seasons. In the case of cassava, there are many papers G x E published from one-year data for fresh and dry weight, starch content, cassava mosaic disease (CMD), cassava green mite and etc.
9) Please improve the methodology about the yield related traits, too low information is provided.
In the current paper, our major focus is fresh root yield and dry matter content. The MM has been edited to serve its purpose.
10) Figure 2: please replace "Main effect" with the corresponding specific trait. In addition, pelase add letters (B) and (D)
Figure 1 has been corrected based on the suggestion
11) Figure 3: the same as Figure 2. Please delete the space symbol
Figure 2 has been amended as per the suggestion
12) the resolution and quality of Figures is too low. Please chnage with at least a 300 dpi resolution
We used the Figure generated via GenStat and the resolution is a bit low. We are happy to any suggestion how to increase the resolution of the pictures
13) References are quite outdated. The scientific literature shows more recent papers. I suggest https://doi.org/10.3390/agronomy13010101 and
https://doi.org/10.3390/agronomy13010101 and other recent publication have been included